# Nanocarriers as a Tool for the Treatment of Colorectal Cancer

**DOI:** 10.3390/pharmaceutics13081321

**Published:** 2021-08-23

**Authors:** Ana Luiza C. de S. L. Oliveira, Timo Schomann, Lioe-Fee de Geus-Oei, Ellen Kapiteijn, Luis J. Cruz, Raimundo Fernandes de Araújo Junior

**Affiliations:** 1Postgraduate Program in Health Science, Federal University of Rio Grande do Norte (UFRN), Natal 59064-720, Brazil; analuiza_oliveira@outlook.com; 2Translational Nanobiomaterials and Imaging (TNI) Group, Radiology Department, Leiden University Medical Center, 9600 Leiden, The Netherlands; T.Schomann@lumc.nl; 3Percuros B.V., 2333 Leiden, The Netherlands; 4Department of Radiology, Leiden University Medical Center, 9600 Leiden, The Netherlands; L.F.de_Geus-Oei@lumc.nl; 5Department of Medical Oncology, Leiden University Medical Center, 9600 Leiden, The Netherlands; h.w.kapiteijn@lumc.nl; 6Cancer and Inflammation Research Laboratory, Department of Morphology, Federal University of Rio Grande do Norte, Natal 59064-741, Brazil

**Keywords:** drug delivery system, nanomedicine, nanoparticles, cancer, biological barriers

## Abstract

Nanotechnology is a promising tool for the treatment of cancer. In the past decades, major steps have been made to bring nanotechnology into the clinic in the form of nanoparticle-based drug delivery systems. The great hope of drug delivery systems is to reduce the side effects of chemotherapeutics while simultaneously increasing the efficiency of the therapy. An increased treatment efficiency would greatly benefit the quality of life as well as the life expectancy of cancer patients. However, besides its many advantages, nanomedicines have to face several challenges and hurdles before they can be used for the effective treatment of tumors. Here, we give an overview of the hallmarks of cancer, especially colorectal cancer, and discuss biological barriers as well as how drug delivery systems can be utilized for the effective treatment of tumors and metastases.

## 1. Introduction

The development of new technologies allows the improvement of several scientific fields, such as the field of medicine, where essential discoveries have been made by applying knowledge and tools that culminate in the improvement of specialized and personalized therapy strategies [1]. The application of improved medical technologies increases the life expectancy as well as the quality of life of patients and, thereby, the standard of life of the general population. Due to technological advances, the equipment and procedures used in the clinic have become less invasive, safer, more effective and optimized. As a result, the patient’s recovery time has decreased along with the risk of complications from surgery, which improves quality of life and reduces the cost of health care [2].

Nanotechnology is a consolidated ally in the search for solutions to as yet unanswered questions regarding some diseases, as well as in the improvement of existing technologies for prevention, diagnosis, control and treatment [3]. The nanomedicine area uses the properties and physical characteristics of materials, structures, devices and systems of a nanoscale size, which approaches the molecular level where changes in biological processes originate and culminate in many pathologies, such as mutated genes, protein defects and infections, among others [4].

With all the molecular knowledge available, it is possible to better understand the pathophysiology of a disease and continuously develop new tools or improve existing ones, which are applied to assist the health of patients. Such improvements are, e.g., sensors and surgical instruments that give detailed information about the location and exact size of a tumor, innovative imaging agents that are specific for a certain tissue, monitoring technologies with improved resolution and sensitivity, and mainly biomaterials with improved targeting characteristics, amongst others [5]. These medical devices are also often used in the diagnosis and treatment of disease and are associated with cell therapy to improve drug delivery, tissue engineering and tissue regeneration. Some technologies integrate multifunctional systems by combining diagnostics, targeted therapy and response tracking, which is an emerging, promising approach that is known as theranostics, a portmanteau derived from the blending of the words therapeutic (thera) and diagnostic (nostic) [6,7].

In the area of drug development, nanomedicine has a prominent role by providing potential solutions to approach challenging medical problems and needs. Nanomedicine applications can provide solutions to diagnostic problems, to therapy monitoring, as well as to the control, prevention and treatment of diseases. Within the therapeutic context, the search for a combination for better treatment efficacy is promoted through the application of innovative molecules, exploring multiple mechanisms of action and maximizing the effectiveness of known drugs, as well as their targeted, accurate and controlled delivery, in order to treat patients more effectively while reducing unwanted adverse effects [8].

## 2. Cancer

Cancer is a disease with genetic changes in DNA, which involves two categories of genes: oncogenes and tumor suppressor genes [9,10]. Oncogenes and tumor suppressor genes are involved in the process of carcinogenesis. Their classification is due to the type of regulation they make in the processes that lead to the promotion of the tumor: (i) if the gene promotes this process, it is called an oncogene, and (ii) if the gene prevents this process, it is called a tumor suppressor gene [11].

### 2.1. Characteristics of Cancer

Hanahan and Weinberg were the first to describe the specific characteristics of tumor cells in the literature [9]. Tumor cells are independent of any growth factor supply. Therefore, avoidance of growth suppressors, prevention of cell death and stimulation of angiogenesis in the environment of tumor cells are observed, and their replicative potential is maintained [9,10]. This list of characteristics was updated later with emerging characteristics (reprogramming of energy metabolism, avoidance of immune destruction) and facilitating characteristics (genomic instability, inflammation) [9,10]. These hallmarks explain the invasion/metastasis of tumors and can be used to find a new approach to diagnosis and treatment [9].

Three of these characteristics of cancer are: preventing cell death, invasion/metastasis and inflammation, which will be discussed in more detail in the next paragraphs.

#### 2.1.1. Preventing Cell Death

The development of resistance to cell death, specifically to apoptosis, is one of the most important factors for the progression and survival of a cancer cell [12]. Apoptosis is a cellular suicide program that leads to the elimination of damaged or abnormal cells during development or after stress mechanisms, and this process involves a series of events that culminate in the activation of initiating caspases (caspase-2, -8, -9 or -10), and these lead to the activation of the executing caspases (caspase-3 or -7) resulting in the cleavage of specific cell substrates in the nucleus and cytoplasm that leads to cell death [12].

There are two ways of signaling cell death due to apoptosis, the extrinsic pathway (related to death receptors) and the intrinsic pathway (mitochondrial; Figure 1) [13]. The extrinsic pathway is initiated in response to the binding and activation of receptors of the tumor necrosis factor (TNF) receptor superfamily; then, there is the coupling of these by the Fas-associated death domain protein (FADD), and this subsequently leads to activation of caspase-8 and formation of a signaling complex on the plasma membrane called the death-inducing signaling complex (DISC), which in turn activates the downstream effector caspases [13]. Caspase-8 also cleaves other substrates, such as Bid, which will stimulate the release of cytochrome c and the activation of the intrinsic pathway [14].

The intrinsic pathway is initiated by the permeabilization of the mitochondrial outer membrane (MOMP—mitochondrial outer membrane permeabilization) and release of apoptogenic factors, such as cytochrome c, apoptosis-inducing factor (AIF) and second mitochondria-derived activator of caspases (SMAC), from the mitochondrial intermembrane space into the cytosol [14]. The release of cytochrome c results in the formation of apoptosomes (cytochrome c + pro-caspase-9 + APAF-1). These complexes promote the activation of caspase-9, which in turn activates effector caspases that result in the execution of apoptosis, while AIF contributes to DNA fragmentation and subsequent chromosomal condensation. Next, SMAC binds to apoptosis inhibitor proteins (IAP), also promoting caspase activation [15].

The main regulators of the intrinsic pathway of apoptosis are the BCL-2 family of proteins that are classified into an antiapoptotic and proapoptotic category, which exist in a competitive flow state. The proportion in a cell can be altered due to the process of cellular stress, since, in harmful situations, BCL-2 of the proapoptotic type called pore-forming executioners are activated and trigger MOMP, which finally results in apoptosis [16].

#### 2.1.2. Invasion/Metastasis

One of the characteristics of cancer is the ability of primary cancer cells to invade adjacent tissues and metastasize elsewhere in the body [17]. This process is complex and requires success in all its stages: from the exit of the primary tumor cells to their transition through the bloodstream, to the escape of the immune system and its stabilization and proliferation at the new location [17]. Studies that aim to understand this process in more detail can help to identify new therapeutic targets to reverse this process, which is the cause of a poor treatment prognosis and high death rates [17].

The expression of some molecules can be used as a metastasis marker. C-X-C chemokine receptor type 4 (CXCR4) is one of these markers of metastasis. Evidence shows that when its ligand, CXCL12, binds to the CXCR4 receptor, it triggers angiogenic properties [18,19]. The prognosis of cancer patients who have an overexpression of CXCR4 in cancer cells is poor, and these patients are at a high risk of recurrence [18].

The chemokine ligand 22 of the C-C motif (CCL22) is a member of the family of chemokines secreted by macrophages and tumor cells and binds to the chemokine [C-C motif] receptor 4 (CCR4) [20]. The interaction between CCL22 and the CCR4 receptor is related to cell proliferation and tumor progression. It is also responsible for attracting regulatory T cells (Treg) to tumor sites, leading to the suppression of antitumor immunity [20]. Based on this, CCL22 levels in serum can be used as a metastasis marker [20].

#### 2.1.3. Inflammation

A wide variety of diseases, including cancer, are mediated by chronic inflammation, which contributes to the maintenance of the inflammatory tumor microenvironment (TME) and predisposes to tumorigenesis [21,22]. Chronic inflammation is characterized by sustained tissue damage and induces cell proliferation and repair. It is associated with all stages related to the formation and promotion process (increased proliferation/survival, activation of angiogenesis and metastasis) of cancer phenotypes [21,22].

Cancer-related inflammation is directly linked to genetic instability: activated leukocytes produce reactive oxygen species (ROS) and reactive nitrogen intermediates (RNI), and inflammatory cells produce cytokines and chemokines that can induce mutations and a neoplastic transformation [23]. In addition, DNA damage itself can also lead to inflammatory processes, such as the release of oncoproteins that can also activate signaling pathways related to this process, in a cycle that culminates in cancerous development [23].

Oxidative stress, a physiological state in which free radicals and ROS are present at high levels, plays an important role in the development of several diseases, including neurodegenerative, metabolic and inflammatory diseases. It has an impact on the initiation and progression of cancer and consequently on its therapy [24].

There are many biomarkers for the measurement of oxidative stress, including the production of lipid peroxidation malondialdehyde (MDA) and the antioxidant tripeptide glutathione (GSH), which can be used to identify different types of cancer. These biomarkers can verify the antioxidant status of patients and thus have great diagnostic and prognostic relevance in oncology, contributing to assessing the most appropriate treatment regime [25,26,27].

### 2.2. Colorectal Cancer

Colorectal cancer (CRC) is one of the most common types of cancer: CRC takes third place in terms of incidence rate and second in terms of highest lethality rate [28]. Men are more affected by this disease than women. Risk factors that increase the prevalence of CRC include smoking, excessive alcohol intake, a sedentary lifestyle, a diet with large amounts of red and processed meat, obesity and having a family history of CRC (Figure 2) [29].

CRC is a slow-developing disease, which is initiated by a single cell that has accumulated genetic and epigenetic mutations over time and, thereby, inactivated tumor suppressor genes and activated oncogenes [30,31]. The sequence of events that culminates in the appearance of the tumor is called carcinogenesis, a complex process that involves modification and mutation in genes that regulate cell growth [32]. Genome instability, named tumor initiation, is part of carcinogenesis, and it can progress to CRC in two ways. The traditional adenoma-carcinoma pathway, which takes place in 70–90% of CRCs, is initiated by a polyp that evolves to an early adenoma. On the other hand, the serration neoplasia pathway takes place in 10–20% of CRCs and evolves from a hyperplastic polyp to a serrated polyp, ending in an adenocarcinoma [30,33].

In the colon, the epithelial renewal of the intestinal mucosa that occurs due to the loss of cells that remain on the tissue surface, and the controlled proliferation that occurs at the base of the crypt where the stem cells are located, are physiological processes. However, mutated cells can divide and reach the colonic lumen, forming discrete adenomas, which over time acquire more mutations and increase in size, developing dysplastic peculiarities that can acquire the capacity to invade other tissues. Thus, the vast majority of tumors in the colorectal region originate from precancerous polyps classified as traditional tubular adenomas, which arise when there is a dysregulation of DNA repair mechanisms, thus causing disordered cell proliferation [34,35,36].

The dysregulation of DNA repair mechanisms occurs due to alterations in genes that regulate cell growth. When mutated, the gene that encodes a tumor suppressor, adenomatous polyposis coli (APC), gives rise to traditional adenomas, while the oncogene BRAF, when mutated, gives rise to serrated polyps [34]. The APC mutation promotes the activation of the Wnt pathway, which leads to the chromosomal instability pathway (CIN), while BRAF mutations promote tumorigenesis through the serrated neoplasm pathway, which leads to microsatellite instability (MSI) [37]. Moreover, not all adenomas will progress to cancer, and carcinogenesis will depend on the accumulation of specific mutations. Furthermore, the time to reach the neoplastic stage will depend on the carcinogenic process and the individual’s exposure to factors that influence progression, such as inflammatory processes. As an example, tumorigenesis via CIN can take 10 years or more, while tumor development via MSI can occur in a few years [38,39].

Despite having similar histopathological characteristics, different CRCs differ in clinical symptoms, response to treatment, molecular characteristics and prognosis [40]. Molecular factors have an influence on prognosis. Patients at a similar stage of the disease at the time of diagnosis may show different treatment responses and different evolutions of the disease due to heterogeneity in their molecular factors [41].

#### 2.2.1. Treatment of Colorectal Cancer

In the past decades, chemotherapy treatment options have advanced from the use of single agents to regimens with a combined use of therapeutic agents. Additional, targeted agents that have improved treatment efficacy in metastatic disease are being used [33,42]. In addition to chemotherapeutic methods for the treatment of CRC, other therapies such as surgery, radiotherapy, targeted therapies, immunotherapy, radiofrequency ablation and palliative treatments are also available.

The choice of treatment depends on several factors, such as the location and tumor stage, which relates to tumor size, tumor growth/infiltration and the presence of locoregional and distant metastases, which are also related to the tumor microenvironment-affected genes, comorbidity and patient prognosis [43]. Thus, if CRC is diagnosed early, there is a greater chance of success in therapy, given that CRC is one of the most treatable cancers when it is discovered in the early stages [30,44].

CRC can be classified into different stages, according to the characteristics of the tumor and the involvement of the nearby lymph nodes, into the TNM classification of malignant tumors (Table 1). This classification is also used to choose the optimal treatment for the cancer patient [45].

In stage I, endoscopic examination of a pedunculated malignant polyp or surgical resection of the tumor and nearby lymph nodes is indicated. Thus, at the time of surgery, it is crucial to examine the number of lymph nodes for the correct staging of the CRC. Malignant polyps are T1 lesions where the cancerous cells have reached the muscle from the mucosa to the submucosa. They may appear benign through endoscopic analysis, but histology should be performed to check for the presence of malignant cells. After the histological analysis, it should be decided whether the endoscopic resection is sufficient or whether it is necessary to execute an endoscopic resection of the mucosa, or other methods, such as endoscopic submucosal dissection or segmental colon resection (performed if the resection margin is less than 2 mm and there is involvement of blood vessels) [46,47,48].

In stage II, the surgical method is performed and the use of adjuvant chemotherapy is not indicated in most cases. According to the European Society of Medical Oncology (ESMO), the treatment using chemotherapy and adjuvants is indicated only when the following criteria are met: little tumor differentiation, vascular invasion, perineural invasion, intestinal obstruction, localized perforation and positive margins. Adjuvant chemotherapy lasts for 3–6 months on one of the following regimens shown in Table 1 [49,50]. In case of stage III, CRC treatment consists of a surgical procedure to remove the tumor, and afterwards adjuvant chemotherapy should be performed, which includes 3–6 months of FOLFOX or CAPOX. In stage IV, the CRC can be controlled by monotherapy or a combination of chemotherapy, targeted biological therapy, immunotherapy, palliative surgery, radiotherapy, and radiofrequency ablation or radio-embolization, as shown in Table 1 [51,52,53].

##### Oxaliplatin

Oxaliplatin (OXA) has been considered an important chemotherapeutic drug and one of the most effective for the treatment of CRC, often used in conjunction with 5-FU/leucovorin or capecitabine, called the FOLFOX or CAPOX regimen; it provides a better response rate and prolonged survival in both treatment metastases and as adjuvant therapy, being considered the gold standard for this condition [54,55,56,57]. It is believed that about 50% of patients with CRC benefit from treatment with OXA [58].

OXA is a third-generation platinum compound characterized by a diaminocyclohexane molecule. It was identified in 1976 and approved for cancer treatment only in 1996, thereby being the first platinum compound to demonstrate efficacy in the treatment of CRC [59]. OXA exerts its cytotoxic activity through direct damage to the tumor cell’s DNA, through the formation of adducts, which induce the formation of DNA lesions, the interruption of DNA replication and transcription, as well as the triggering of immunological reactions that culminate in apoptosis and cell death [60,61].

As with other chemotherapeutic agents, it is known that the use of OXA is associated with certain limitations such as systemic toxicity and innate and acquired resistance, which are related to an increase in patients’ morbidity and mortality [62]. This drug has a wide spectrum of adverse effects that affect the most diverse organs and can cause anemia and cytopenia; nausea, vomiting and diarrhea; mucositis and stomatitis; hearing loss; hepatic, cardiac and renal dysfunctions; neurosensory toxicity and neuropathies; as well as alopecia, anorexia and asthenia [46,47,63,64].

Resistance of tumor cells to OXA, which is one of the main reasons for treatment failure, can be caused by increased DNA repair mechanisms and increased expression of drug efflux pumps (Figure 3), as well as by genetic mutations in target pathways (such as folate, VEGF, EGF and microsatellite instability). However, these factors can be used as predictive markers for the response of the tumor cells to therapy [58].

##### Other Treatments

Radiotherapy, as well as other treatments, depends on the clinical stage of the tumor and the age of the patient. This treatment is mainly administered in patients with an advanced stage of cancer, i.e., III or IV or even nonmetastatic CRC with positive lymph nodes [48]. Several methods are used for this purpose, such as short-term radiation therapy, which has been a popular method applied in Europe for resectable rectal cancer and also in patients with locally invasive tumors [30,51].

In addition to the aforementioned treatments, palliative treatment emerges as a crucial component of standard cancer care for patients with stage IV disease, and, more recently, it has also been included in the treatment of earlier cancer stages. Palliative treatments include: combination of surgery, combined chemotherapy/targeted therapy, immunotherapy and radiotherapy regimens [52,53]. Palliative surgery is the most used for cases where patients have symptoms such as obstruction, perforation or bleeding [65]. Hence, palliative therapy for advanced CRC aims to prolong the patient’s survival, providing greater quality of life and lower costs in health care [45,65].

Taken together, the choice of the approach used in the current treatments is based on the stage of evolution and progression of CRC, considering the occurrence of invasion and/or metastasis in a standardized way for all cancer patients. However, the incorporation of the use of nanoparticles (NPs) as a treatment in CRC could make the treatment more specific to the needs of each patient. The use of these NPs as a drug delivery system (DDS) may help to overcome the biological barriers that hinder conventional treatment [66,67].

## 3. Biological Barriers

Biological barriers are major difficulties to overcome in the use of nanosystems for the treatment of cancer patients. The unique interactions between NPs and the biological components of the body are intrinsic to each patient, making it difficult to systemically apply nanomedicine in cancer [68,69].

### 3.1. Reticuloendothelial System

One of the biological barriers to overcome is the mononuclear phagocyte system (MPS). This is one of the constituents of the immune system and is formed by phagocytic cells and macromolecules that can interact with NPs and trigger an immune response, which makes it difficult for nanocarriers to enter cells [70]. To escape the MPS, the physicochemical properties of the NP’s surface are a key factor, which may increase their time in the blood. A good example are NPs coated with polyethylene glycol (PEG), which are able to escape the MPS, thus increasing the possibility of the drug reaching its site of action [71].

These physical and chemical properties also affect which components of the MPS will act on the NPs, as, for example, a hydrophobic surface of NPs allows them to be normally captured by the liver, spleen and later by the lungs. NPs with a hydrophilic surface have less absorption by the liver and spleen [72]. The size of the NPs is another characteristic that determines the recognition by the MPS, which has been researched. NPs bigger than 250 nm are removed by the MPS, opsonized and captured by macrophages [73]. Even if the modification of the surface reduces recognition by the MPS, complete evasion has not yet been achieved [71].

### 3.2. Renal System

Another difficulty when using NPs in vivo is the renal system, which filters the circulating blood, produces hormones and promotes homeostasis of the body [74]. When formulating NPs, the physicochemical properties must be taken into account so as not to affect the rate of renal clearance. Research has observed that a change in size, even if it is only by 2 nm (from 6 to 8 nm), can affect kidney clearance, because the smaller the size, the higher the clearance rate [71,75].

However, changing the size of NPs to prevent or decelerate renal clearance can affect the performance of NPs. One way to solve this problem would be the formulation of biodegradable nanosystems, which could be eliminated more easily by the kidneys. However, they can alter the delivery of the drug by altering the surface load [71,76]. Thus, before formulating the DDS, one must choose which are the main benefits that are necessary for their objective [71,76].

### 3.3. Blood-Brain Barrier

The blood-brain barrier (BBB) is a structure that protects the brain. As a consequence, blood vessels of the BBB only allow specific molecules to pass through, to allow nutrition of the central nervous system and to prevent the entry of harmful agents [77]. Normal BBB is formed by brain capillary endothelial cells and is organized as a barrier that surrounds brain capillaries. It has specific transporters that deliver essential biomolecules to the brain, while larger molecules cross the barrier through receptor-mediated transcytosis [77].

On the other hand, the BBB can be an obstacle for the systemic treatment of diseases, such as cancer, in the central nervous system. During progression, brain tumors alter the BBB, which then becomes the blood-tumor barrier (BTB), being quite heterogeneous and consisting of capillaries with the following characteristics: (i) continuous and without perforations (such as the vasculature of a healthy brain), (ii) continuous and having perforations (changes permeability) and (iii) noncontinuous (with interendothelial gaps). Even though it is more permeable to smaller and larger molecules, the BTB still possesses BBB characteristics, making it one of the biggest obstacles in the treatment of brain tumors and metastases [78,79].

Brain tumors or brain metastasis have high rates of tumor recurrence, even after initial surgical interventions and pharmacological treatments, since the delivery of pharmaceutical active agents to invasive cells behind the BBB is a challenge. To increase BBB permeability in cancer patients, current treatments consist of direct, intraventricular or intracerebral injections, infusion and even implantation. However, these therapies are toxic, can cause infections and can improperly release medications, which can lead to the accumulation of the drug in the ventricles, subarachnoid space, cerebrospinal fluid or blood [80,81].

NPs seem to be a good method to overcome the BBB, since the functionalization of their surface and their size can be changed [80]. Gold NPs (AuNPs) have been investigated for their advantages, such as stability, ability to interact with light, synthesis of different sizes, and functionalization with peptides, proteins and other biomolecules. Additionally, they can be visualized easily with computed tomography (CT), a noninvasive procedure for patients [82].

### 3.4. Pathophysiological Barriers in Cancer

Finally, pathophysiological barriers are another challenge for nanoformulations. The enhanced permeability and retention (ERP) effect, attributed to the deficiency of lymphatic vessels in the tumor, causes the intratumoral accumulation of NPs [76,83]. The tumor-associated angiogenesis process leads to the formation of irregular vessels, with more gaps, promoting an increase in factors such as nitric oxide, VEGF, tumor necrosis factor (TNF) and bradykinin, making it difficult for NPs to enter and act. The proliferation of tumor cells leads to stress in the TME, causing compression of the vessels, making it difficult for NPs to penetrate the tumor tissue [76,84].

In addition to the physicochemical properties of NPs, already-mentioned characteristics of the TME, such as the ERP effect and the pressure of tumor cells, are the factors that influence the entry of NPs into the tumor. Furthermore, the location of the tumor, the type of cancer, the stage of cancer and other factors intrinsic to each patient affect the penetration of NPs [85]. Therefore, measures must be taken to facilitate the penetration of NPs into tumors, such as previous treatments to reduce the pressure of the TME, as well as the inhibition of signaling by VEGF and TGFβ, thus minimizing the stress of the TME and increasing entry of the NPs into the tumor [76].

## 4. Nanoparticles as Drug Delivery Systems

Among nanomedicine projects, studies on NPs that are used for the purpose of drug delivery are one of the most widespread. NPs are complex molecules, composed of superficial layers, which can be functionalized with a variety of elements (such as ions and surfactants), cores (different chemical materials) and nuclei (central portion) [86]. Several methods can be used for their synthesis, which can be classified into either the destructive approaches starting from a larger molecule, called “top-down”, or the constructive ones, called “bottom-up”. They can also be characterized with different morphological, structural and optical techniques in order to thus obtain control of their reactivity, resistance and other properties [86].

The formulation of NPs, which can be used as DDSs, occurs through the association of active drugs with colloidal nanostructures of a size smaller than a micrometer and a large surface area in relation to their volume. The type of material used for these nanostructures can include organic (such as polymers, dendrimers, solid lipids, micelles and liposomes), inorganic (such as silica, gold, quantum dots, carbon NPs and nanotubes) and hybrid nanocarriers, which combine the advantages of both materials (Figure 4) [87].

Polymer-lipid hybrid NPs (PLHNPs) are an example of hybrid nanocarriers for cancer therapy; this nanosystem has a polymeric core where the drug is encapsulated, surrounded by a lipid layer. The combination of polymeric NPs and liposomes increases the biodistribution of the system, in addition to overcoming the limitations that each system would have alone [88,89,90]. Another example of a hybrid nanocarrier is mesoporous silica NPs (MSNs) covalently bonded with poly (oligo(ethylene glycol) monomethyl ether methacrylate) (POEGMA) and targeting peptide (RDG) to perform system stabilization and deliver the drug 5-fluorouracil (5-FU) in a targeted and effective manner. This hybrid nanocarrier has a better efficacy against CRC than the free drug, thus showing the importance of hybrid nanosystems and how they can be efficient in cancer therapy [91].

Among the organic NPs are the micelles, which are spherical macromolecules formed by a hydrophilic outer layer and a hydrophobic interior in an aqueous solution [92]. This type of nanocarrier is thermodynamically stable as well as considered nontoxic and safe due to its composition of biocompatible material, being able to transport hydrophobic drugs, in vivo, inside or on the surface through a covalent connection with the carrier [92]. Its small diameter allows for the gradual administration of drugs and prevents their recognition by the immune system and their filtration by the spleen epithelial cells. In addition, the micelles can be modified in order to alter their functionalization, thus promoting the delivery of the drug and prolonging the therapeutic effect of the drug [92]. As an example of micellar nanoparticles, researchers recently formulated micellar NPs targeting small-sized trastuzumab by removing a surfactant that produced concentrated phthalocyanines with strong near-infrared absorption. In combating tumor lymph node metastases in CRC in vitro and in vivo, targeted micellar phthalocyanine (T-MP) showed promising results [93].

Liposomes are spherical lipid vesicles composed of a lipid outer layer and an aqueous interior, with variable sizes ranging from nm to µm [94,95]. They are biocompatible and stable colloidal macromolecules. Their structure, being very specific, allows for the fusion of membranes with the target cell. Liposomes can encapsulate hydrophobic drugs, promoting solubilization in water and prolonging the release of drugs that have a short half-life, which is of great importance for cancer therapy and other diseases [94,96]. Moreover, the functionalization of liposomes is advantageous: PEGylated liposomes have a longer half-life in the circulation, and various other ligands (such as proteins, carbohydrates and peptides) can also be used on the surface of the liposome in order to reach the tumor site [97]. A recent study formulated and characterized liposomes containing irinotecan (IRI) to treat CRC; this research achieved a reduction in prostaglandins in colonic tissue in addition to a reduction in tumors in mice, demonstrating promising results with these vesicles [98].

In addition to the abovementioned systems, the most-used system for delivering and targeting anticancer drugs are polymeric NPs. They are solid colloidal systems formed by the polymerization of monomers or polymers that surround the drug [99]. Their structure can vary in size and molecular weight depending on the method of preparation. This variation in structure can produce nanocapsules, where the drug is confined in an aqueous cavity and is surrounded by a single polymeric membrane or nanospheres, in which the drug is dispersed within the particles [100]. The formulation of nanocapsules and nanospheres is complex. Nevertheless, these NPs have many advantages: they are biocompatible, stable, have a nontoxic nature and large-scale production is feasible [101].

Inorganic NPs are also widely studied as agents in antitumor therapy. Among them are AuNPs that have versatile physical and chemical properties, such as surface plasmon resonance (SPR), a large surface-to-volume ratio, fluorescence quenching and the ability to form stable chemical bonds with groups containing sulfur and nitrogen [102]. AuNPs are effective for medical imaging purposes and medication administration, due to their shape and diameter (1 to 150 nm) [103]. This nanosystem contains a large surface area that can be adjusted for drug loading, conjugation or binding of genetic material, thus enabling drug solubility, stability and more useful pharmacokinetic parameters [103,104]. Studies have shown that AuNPs reduce the population of tumor-associated fibroblasts and type I collagen production in CRC. In addition, AuNPs decrease vascular endothelial growth factor (VEGF) signals via the Akt signaling pathway, thereby reducing the pressure applied by the tumor and increasing vascular permeability (reviewed in [92]). A disadvantage of using these NPs must be noted: the possible toxicity, since recent research has shown that smaller AuNPs can accumulate in organs, such as the brain, kidney, liver, spleen and lung, and can be internalized by cells, promoting more cellular toxicity [105,106]. AuNP systems containing cetuximab (cetuximab-AuNPs) have shown promising results in in vitro studies against CRC. Cetuximab is a monoclonal antibody against the epidermal growth factor receptor (EGFR), and it has been widely investigated since EGFR is overexpressed in CRC and plays important roles in tumor survival. Thus, when blocked, EGFR would reduce damage caused by tumor cells. Results showed the cytotoxicity of cetuximab-AuNPs to cancer cells, with NPs mainly possessing a size of 60 nm. It was also possible to analyze the change in the expression of biomarkers on the surface of the cancer cell, where, after treatment with cetuximab-AuNPs, there was a greater expression of epithelial cell markers: epithelial cell adhesion molecule (EpCAM), melanoma cell adhesion molecule (MCAM) and human epidermal growth factor receptor-3 [107].

Silica NPs are studied for targeting drugs to tumors, as they are more advantageous than other inorganic NPs due to their ability to release drugs and their biodegradability [97]. Within the category of silica NPs, there are mesoporous silica NPs (MSNs) that are attractive due to their physicochemical properties: adjustment of the particle size (about 10 nm), adjustment of the pore size between 2 to 50 nm (according to the shape and size of the drug) and greater surface area linked to low cytotoxicity [97]. In addition, MSNs have a rich surface of silanol groups, which can be changed with molecules and functional groups such as polymers, metals, metal oxides and targeting binders, among others, for the final function of MSNs [97]. Anti-miR-155-loaded MSNs modified with polymerized dopamine (PDA) and AS1411 aptamer (MSNs-anti-miR-155@PDA-Apt) were used in a study to assess their therapeutic potential against CRC. miR-155 is expressed at high levels in the CRC TME, so the researchers studied miR-155 and its correlation with NF-kB. The results showed that there was a correlation between NF-kB and miR-155, and that NP MSNs-anti-miR-155@PDA-Apt were promising for the treatment of CRC [108].

The physicochemical properties of the material can also be adjusted by modifying its compositions, dimensions, shapes and surfaces, thereby creating more effective, biodegradable, biocompatible, targeted and responsive products [109]. Currently, a considerable number of potential drugs and those that are already in use would benefit from improvements in their pharmacokinetics and biopharmaceutical properties [110]. The NPs can carry drugs by various methods, such as encapsulation and surface fixation. They efficiently penetrate through barriers, such as cell membranes, and deliver the drug to the target site [111].

The use of DDSs is a strategy that has been developed and widely investigated to improve the transport of drugs to the site of action in target cells or tissues. More specifically, it is defined by the enhancement of bioavailability through nanoengineering, which improves the pharmacological and therapeutic properties of the DDSs [112]. DDSs have also become relevant with regard to the: (i) solubility of hydrophobic drugs; (ii) reduction of systemic toxicity, enzymatic degradation and side effects; (iii) creation of absorption and targeting mechanisms; (iv) increased specificity and efficiency; (v) continued release to maintain the therapeutic dose; and (vi) structure of various special applications such as ocular, neurological and anticancer therapy [113,114].

Some characteristics are desirable for the design of biological NPs, such as: (i) chemical compatibility with physiological solutions; (ii) ease in designing and modifying; (iii) natural composition of material; and (iv) biocompatible, biodegradable and noncytotoxic nature [115]. When used as a strategy to fight cancer, the NPs used as DDS have an additional feature, besides advancing the bioavailability and solubility of drugs. Tumor cell-specific binding systems can be added to the surface of NPs, which protects healthy tissues, resulting in decreased cytotoxicity [67,116].

### 4.1. Poly-Lactic-Co-Glycolic Acid

The effectiveness of NPs, organic ones in particular (liposomes, micelles, polymers, dendrimers), is remarkable due to the biocompatibility and biodegradation of these systems [117]. Thus, polymers stand out in the delivery of drugs due to their formulation, stability, longer half-life and nontoxic nature [117]. Polymeric NPs consist of nanospheres or nanocapsules, and the preparation methods of such nanosystems vary depending on the drug to be encapsulated and the route of administration. However, there are two general methods of preparing polymeric NPs: the “top-down” and the “bottom-up” methods, and both methods used generate products that are obtained with aqueous colloidal suspensions. The steps of the technique that are used in the “top-down” method include: emulsion evaporation, emulsion diffusion, coacervation and nanoprecipitation. The “bottom-up” method steps include: emulsion polymerization, interfacial polymerization, interfacial polycondensation and molecular inclusion. Thus, both use organic solvents to dissolve the polymer. Therefore, the solvent must be removed in subsequent reactions to avoid compound toxicity [118,119].

Polymers are classified by their form of extraction, which can be natural or synthetic. Among the natural ones, the most studied are: chitosan, alginate, dextran, and polymers such as pullulan and hyaluronic acid (HA). Among the synthetic polymers, polylactic acid (PLA), poly-ε-caprolactone (PCL), PEG and poly-lactic-co-glycolic acid (PLGA) are the most frequently studied [120].

PLGA, also known as “smart polymer”, is a biodegradable synthetic polymer that started to be used in the early 1970s in the composition of absorbable filaments of surgical articles [121]. Its application has been extended in the last decades, becoming one of the most successful DDSs due to its remarkable properties: its biocompatibility, sustained release, nontoxicity, nonimmunogenicity, easy adaptation of the polymer to various types of drugs and hydrolysis result in endogenous and easily metabolized compounds [122,123,124].

Regarding its physicochemical properties, PLGA is a linear aliphatic copolymer made of lactic acid and glycolic acid monomers in different proportions. PLGA is quite adaptable and can be processed in completely amorphous or highly crystalline forms, in almost any shape and size, being able to encapsulate the most diverse molecules (both hydrophobic and hydrophilic) [125,126]. In addition, PLGA is soluble in a wide range of common solvents, including chlorinated solvents, tetrahydrofuran and ethyl acetate [127]. All these and other characteristics can still be modulated by structural changes, which makes PLGA suitable for various biomedical devices and which promotes improvements in drug stability, degradation, release and targeting [125].

In the clinic, PLGA NPs have already been used as DDSs in various pathological conditions, such as different types of tumors [128,129,130,131,132], bone metastasis [133], tuberculosis [134], leishmaniasis [135,136], fungal infections [137], bacterial infections [138,139], atherosclerosis [140], inflammation [141,142], cystic fibrosis [143] and glaucoma [144]. An example of PLGA NPs investigated as therapy for CRC are the PLGA nanoparticles encapsulated in α-mangostine (Mang-NPs). These nanoparticles have demonstrated the ability to inhibit the viability of CRC cells, epithelial-mesenchymal transition, formation of colonies and to induce apoptosis. Mang-NPs were also able to inhibit the signaling of the Notch pathway by reducing the expression of Notch receptors (Notch1 and Notch2). Thus, polymer-based Mang-NP nanoparticles are potential therapeutic targets, demonstrating the potential of polymers in CRC therapy [145].

### 4.2. Properties of Nanocarriers

When formulating NPs, some characteristics need to be ideal in order to use them as a DDS, such as size, shape, solubility, surface charge and targeting capacities. These characteristics must be controlled during the formulation process of the NPs and can be measured at the end [67].

#### 4.2.1. Physicochemical Properties

With regard to size, two factors must be taken into account and balanced: (i) internalization by cells and (ii) clearance from the system of the organism in vivo [67]. The smaller the NPs, the greater the levels of cellular internalization. However, NPs smaller than 10 nm will be more easily eliminated by renal clearance. In addition, larger NPs (> 200 nm) are eliminated from the bloodstream more quickly, by the complement system, and will accumulate in the liver and spleen [146].

In addition to the size, the shape of the NPs affects the surface-area-to-volume ratio, which changes the NPs’ pharmacokinetic properties, cell internalization and toxicity. Thereby, the shape influences the effectiveness of the DDS [147,148]. The shape of NPs is crucial to determine the drug’s destination in vivo, as the shape influences the permanence in the bloodstream, the uptake of the NPs by macrophages and the biodistribution [149]. Currently, most of the nanocarriers studied are spherical, but nonspherical nanocarriers are promising because they are hydrodynamic, spending more time circulating. The shape of the NPs interferes with biodistribution: aspheric nanocarriers accumulate less in the liver than spherical ones [149,150].

Another important feature is the surface chemistry, which determines the surface charge, hydrophobia and ligands, among others. This parameter plays a fundamental role in the interaction of the nanocarrier system with the biological microenvironment, as it influences colloidal behavior, interactions with plasma proteins and transmembrane permeability, thus being able to change the direction of the nanocarriers in vivo [151].

In the case of nanocarriers in cancer therapy, the surface charge is important. Cells in the human body have a predominantly negative charge on their surface (−40 to −80 mV). However, the surface of tumor cells is even more negatively charged, since the cell membrane contains more negative phospholipids, such as chorionic gonadotropin, anionic RNA residues and sialic acid. In contrast, the surfaces of normal cells in the human body have higher amounts of phospholipids with a neutral charge [152]. Nanosystems with high positive charges are nonspecific and can interact with the surface of blood vessels, being internalized by cells and eliminated from the blood circulation. Therefore, the surface charge must be analyzed in order not to interact with other cellular components in the circulation and thus be able to reach the TME [153,154].

Another important characteristic of NPs is their stability in solution, which is reflected by the absolute value of the zeta potential. Zeta potential values between 0 mV and ± 5 mV are not ideal, as they indicate the high instability of the NPs and consequently a rapid aggregation, whereas values greater than ± 30 mV are desirable and indicate the stability of the colloidal system [155]. Such a desirable zeta potential, which will ensure the stability of the system, can be acquired by modifying the surface of the particle through the addition of a coating. Various coatings can be used for this purpose, such as acrylate, bovine serum albumin (BSA), citrate, N-acetyl cysteine, polyacrylic acid (PAA), PEG and polyvinyl pyrrolidone (PVP) [155].

#### 4.2.2. Solubility, Degradation and Clearance

The ability to solubilize a drug is one of the many advantages of nanomedicine. Many drugs used to treat cancer are insoluble in water and require the administration of aggressive solvents. As an example, the use of paclitaxel requires polyethoxylated castor oil, and, to avoid side effects of the solvent, corticosteroids and antihistamines are administered to patients [156,157]. Importantly, due to its benefits, Abraxane^®^, the albumin-based NP containing paclitaxel, has been approved by the FDA as a therapy against various types of cancer. It causes less neutropenia and has a lower risk of hypersensitivity reactions [158,159].

To be successful, NPs must also be formulated with the ability to escape from phagocytic cells and endolysosomes in order to reach the cytosol of tumor cells. In this way, different nanocarriers have been created to reach the site of action [160]. A good example is the coating of NPs with polymers containing amine groups (such as PLGA); these polymers are able to evade endosomes due to the “proton-sponge effect” [160]. Nanosystems that use the “proton-sponge effect” contain cations on their surface, which sequester protons through the proton pump, thereby increasing the amount of water molecules within the endosomes, leading to the swelling and subsequent rupture of the endosomal vesicle [160].

In addition to escaping degradation, nanocarriers must be formulated in order to avoid rapid clearance in the bloodstream. The main system that carries out the elimination of nanosystems is the MPS, also known as the reticuloendothelial system (RES), which is composed of the liver (Kupffer cells), spleen and bone marrow. This system retains and eliminates NPs [161].

Nanosystems without surface changes are quickly recognized by macrophages of the MPS. One way to avoid this phenomenon is to add nonionic polymers or surfactants to the surface of the system, which allows a longer circulation of NPs [162,163]. PEG is a polymer that has exceptional physical properties, such as water solubility, total biodegradability and a high degree of conformational entropy, being ideal on the surface of NPs to improve stability and prolong the circulation of the nanocarrier [162,164]. Thereby, PEG increases the systemic circulation time of the NPs [165].

#### 4.2.3. Targeting

Nanomedicine has allowed an improvement in traditional therapies, which would normally be swiftly cleared from and are widely distributed throughout the body [166]. The use of targets, specific for tumor cells, through their addition on the surface of NPs, allows the drug to be delivered to a specific tissue [167]. This targeted DDS increases the efficiency of the drug by accumulating in the target tissue and decreases the drug’s side effects elsewhere in the body [166]. For the targeting of drugs, there are two basic strategies: passive or active targeting. Passive targeting is based on the physical and chemical properties of the NPs and their retention at the disease site, using blood irrigation and vessel permeability. Active targeting, on the other hand, depends on an interaction between a ligand on the surface of the NPs and exclusive receptors on the target cells, providing a better performance by increasing the uptake of the NPs and thereby the effectiveness of the drug [168]. There are several types of moieties, ranging from small organic compounds, such as trisodium citrate, to large polymers, such as polyethylene glycols. In addition, functional biomolecules, such as proteins, hyaluronic acid, folic acid (FA), carbohydrates/polysaccharides, lipids, antibodies and oligonucleotides, can be applied as targets (Figure 5) [169,170].

The use of antibodies on the surface of NPs to target tumor cells is a promising strategy for their specific high-affinity binding to target cells. An example is the use of cetuximab or matuzumab on the surface of liposomes for drug delivery in CRC [171]. This antibody binds to the EGFR, which is overexpressed in several types of cancer such as colorectal carcinoma. The antibody promotes the inhibition of EGFR signaling, thereby reducing a factor that is responsible for tumor development and progression [172,173].

Carbohydrates, such as galactose, are a category that is widely used to target NPs. This molecule binds to the asialoglycoprotein (ASGP) receptor, found in abundance in liver tumor cells [174]. Research carried out shows that the galactose-lithocholic acid-PEG-lactobionic acid nanosystem that carries doxorubicin (DOX) increases the internalization of NPs by human liver cancer cells (SK-HEP-1), promoting the death of tumor cells and reducing tumor growth when compared to untargeted NPs [175].

Another molecule that has been widely studied for its characteristics in directing the delivery of NPs is HA, a natural polysaccharide present in the human body which constitutes the extracellular matrix and is biodegradable, chemically modifiable and hydrophilic, drawing attention in the field of nanomedicine [176,177]. In addition, its CD44 receptor is overexpressed on the surface of the tumor cells of many types of cancer, and its interaction with the ligand is related to tumor progression, infiltration and metastasis [178]. Thus, formulations of NPs containing HA have been of major interest for cancer therapy. As an example, micelles formulated with HA-PLGA containing DOX demonstrated an increased uptake and greater cytotoxicity for human colon cancer cells when compared to a system without HA [179].

One of the most used biomolecules as a target in NPs is cholesterol (CHOL), an indispensable structural component of cell membranes. CHOL has attracted attention in recent years for providing structural and functional improvements in DDSs with regard to their size, drug load, encapsulation efficiency, hydrophobicity, biocompatibility and biodegradability [180].

It is also important to highlight the role of CHOL in tumor cells, since tumor cells require more nutrients due to their high rate of proliferation [181]. Consequently, more CHOL is needed for the formation of new cell membranes and to maintain tumor progression. Therefore, conjugation with CHOL selectively targets the NPs for tumor cells and allows for an increased drug concentration with a desired therapeutic effect inside the cell [181]. This active targeting will facilitate the capture of NPs in cancer cells via endocytosis [182]. Since tumor cells, compared to normal cells, require higher amounts of CHOL due to their high metabolism, this will increase the internalization of CHOL and, consequently, the internalization of the NPs as well as the drug they carry [181,182].

An alternative to improve targeting and to reduce side effects are FA ligands, which recently gained popularity. FA is known for its high affinity to folate receptors, which are often overexpressed in tumor cells, thereby promoting the selective uptake of NPs. In addition, FA is an important vitamin for DNA biosynthesis and is widely consumed by proliferating cells [183,184]. Folate receptors are overexpressed in, for example, colorectal, lung, breast, ovarian and endometrium cancers [183]. The properties of FA, such as stability, nonimmunogenicity, simple conjugation chemistry and rapid internalization, are advantageous over other ligands. In addition, this vitamin improves cytotoxic and apoptotic activity. Therefore, FA has the potential to be used in many types of cancer treatments [185,186,187].

### 4.3. Application of Nanoparticles As Drug Delivery Systems for Cancer Treatment

Current cancer treatment approaches face challenges regarding disease control and patient survival, which result from a failure to control metastasis, promotion of systemic toxicity, adverse effects and drug resistance, and subsequent death [188,189,190]. The characteristics of the tumor, such as the fact that the molecular environment of solid tumors is unstable due to the accumulation of several genetic mutations over successive cell cycles, also contribute to treatment failure [191]. The genetic variability of the tumor depends on the tumor type, the tumor growth site, the intrinsic characteristics of each patient and the occurrence of metastases, among others [192,193]. Due to this variability, the success rate related to the treatment of solid-tumor cancers is still minimal. Therapeutic failures of medications are often connected to difficulties regarding an efficient direction to the tumor sites, i.e., targeting, and their ability to penetrate physical barriers. The immunosuppressive TME is also a barrier to the chemical recognition of drug molecules and promotes genetic changes in cancer cells, leading to drug tolerance [194,195].

Characteristics that are intrinsic to solid tumors make it difficult to achieve therapeutic efficacy. These characteristics are: the histopathological structure, the insufficiency of specific antigens and an immunosuppressive microenvironment. The stroma of solid tumors develops a physical barrier that inhibits and regulates the movement of fluids, gases and cells, and which is also able to limit the influx of drug molecules [196]. The tumor stroma, in addition to acting as a structural support for the development of the tumor, also contributes to new genetic and molecular changes [196]. Moreover, the tumor’s immunosuppressive microenvironment is a barrier to the action of chemotherapeutics, promoting genetic changes in cancer cells that lead to drug tolerance [197]. These factors make it difficult for the drugs to penetrate the tumor mass, thereby promoting a delay in the patient’s clinical responses and even disease progression [198].

In order to improve antitumor therapy, several studies based on targeted or personalized therapies and even immunotherapy trials are being developed [195]. These studies resulted in the identification of molecules expressed by tumor cells as well as released growth factors, proteins, chemokines and cytokines that were associated with tumors, in order to develop individualized treatment [195,199,200].

Targeted or personalized therapy can be achieved through the use of DDSs when chemotherapy is incorporated into NPs and specific targeting molecules are added to the surface. As a consequence, the targeting of the therapeutic compound to the specific tumor site can promote therapeutic efficacy and reduce adverse effects. Adverse effects would be caused by chemotherapy alone, due to a lack of specificity and selectivity for the tumor sites during systemic administration [199,200]. In addition, DDSs have a longer circulation time due to their favorable pharmacokinetics, which delay elimination and/or excretion. Prolonged circulation results from the nanometric size of the DDSs as well as their composition containing tumor site-specific targeting moieties [201,202].

The TME strongly promotes angiogenesis, which leads to leaky spacings in the endothelial vessels and, consequently, a high vascular permeability. The space between tumor vascular cells depends on the type of tumor, ranging from 100 nm to 800 nm, which allows molecules within this size range to reach the tumor tissue [160]. In tumor blood vessels, there is also a significant absence of pericyte cells, which are important for the stabilization and support of blood vessels as well as smooth muscle cell layers. These structural changes cause a high vascular pressure, with a larger influx of molecules [160]. In addition, these factors, which are associated with the reduction of lymphatic drainage in tumors, ensure that the nanometric agents preferentially reach the tumor site and, due to their size, accumulate in this tissue [203,204,205]. This process is known as the EPR effect. Therefore, the size of the DDSs, their prolonged circulation and the characteristics of the tumor vessels allow for the majority of the nanometric agents to accumulate in the tumor tissue and release their therapeutic content in a target-specific manner, thereby avoiding side effects [206].

The search for better treatments has led to NPs being used as DDSs and diagnostic tools as well as for the molecular imaging of gene delivery approaches [207]. Several formulations have already been approved for use in the clinic by the US Food and Drug Administration (FDA) or the European Medicines Agency (EMA). Once approved, NPs have proven their safety and efficacy in humans and, if commercialized, are likely to meet standards of good manufacturing practice [207]. The first approved formulation was Doxil^®^, a liposome used to deliver DOX, and the most recently approved NP is Apealea^®^, a micelle system used to deliver Paclitaxel (PTX) [207]. For Doxil^®^, these approvals by the FDA and EMA took place in 1995 and 1996, respectively, and Apealea^®^ was approved by the EMA in 2018 [158,207]. Approvals were granted for several types of systems, such as liposomes, polymeric micelles, albumin or inorganic NPs [207]. The design characteristics of these already approved nanomedicines are similar, such as the inclusion of PEGylated or non-PEGylated structures, encapsulating a single drug [207].

Some of these formulations are already available for use in the clinic, while others are in Phase II/III clinical trials; most of these are used for intravenous administration, but some have been developed for intratumoral administration [207]. Among these, several nanomedicines are being tested in solid cancers and CRC. A cyclodextrin-based NP-camptothecin conjugate is currently in clinical trials for the treatment of CRC. This conjugate NP will be used for theranostics via intravenous administration but has not yet been clinically approved [207].

Taken together, nanosystems need to meet several of the above-mentioned properties at the same time to be suitable as efficient DDSs.

## 5. Conclusions

Novel theranostic tools provided by cutting-edge nanotechnology could be helpful in overcoming the challenge faced in the treatment of patients with cancer. This review presents an overview of state-of-the-art nanomedicine-based cancer therapies, with special attention paid to CRC. This review does not only explore alternatives for the treatment of tumors by means of NPs but also highlights several strategies to hurdle these biological barriers which prevent effective treatment.

The biological barriers discussed in this review include the reticuloendothelial system, the renal system, the blood-brain barrier and several other pathophysiological barriers in cancer. Once these barriers are overcome, nanoformulations benefit from the ERP effect of the TME, which results in intratumoral accumulation and treatment.

Nevertheless, factors such as the location of the tumor, tumor characteristics and the TME are different for each patient and affect the penetration of the DDS. Therefore, future nanotechnologies for drug delivery need to be tailored to tumor biology, which will significantly increase the efficacy of the treatment and, at the same time, reduce the side effects of anticancer drugs that arise from systemic administration. Therefore, the application of novel nanotechnologies increases the quality of life of patients and their life expectancy in general.

## Figures and Tables

**Figure 1 pharmaceutics-13-01321-f001:**
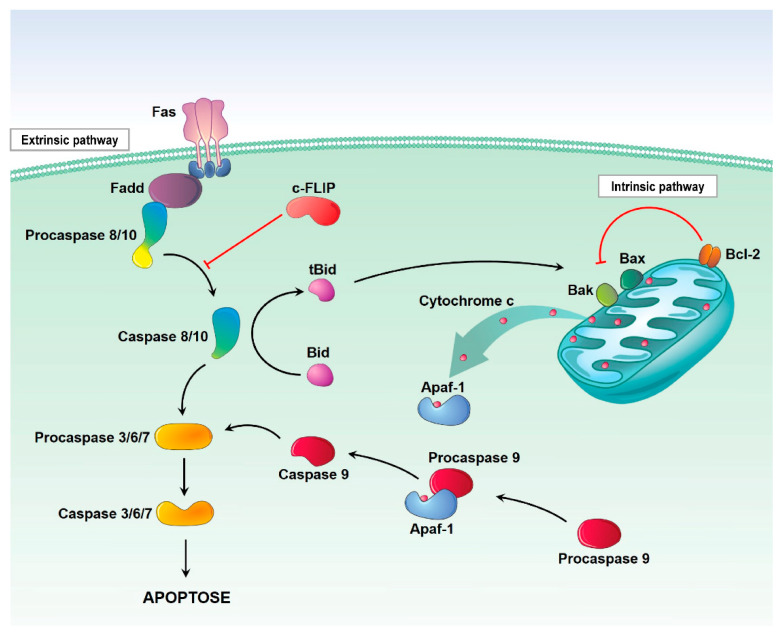
Pathways of apoptosis. Apoptosis can occur through two pathways: The extrinsic or the intrinsic pathway. The extrinsic pathway begins with the binding of extracellular ligands to death-promoting receptors, through the Fas-associated death domain adapter protein (FADD), then recruits procaspase-8, which in turn activates caspase-8. The intrinsic (or mitochondrial) pathway is regulated by a series of specific death-promoting molecules released from the mitochondria. Members of the pro- and antiapoptotic BCL-2 family compete on the surface of the mitochondria to control the release of cytochrome c. The released cytochrome c is associated with Apoptotic protease activating factor-1 (Apaf-1) and procaspase-9, which activate caspase-9. The two pathways share a common end point at the level of caspase-3 activation. The interaction between these pathways occurs through the Bid cleavage triggered by caspase-8. Bid’s interaction with Bax or Bak in the outer mitochondrial membrane results in the release of cytochrome c. The antiapoptotic BCL-2 family can prevent such a release by direct interaction with Bax and/or Bak.

**Figure 2 pharmaceutics-13-01321-f002:**
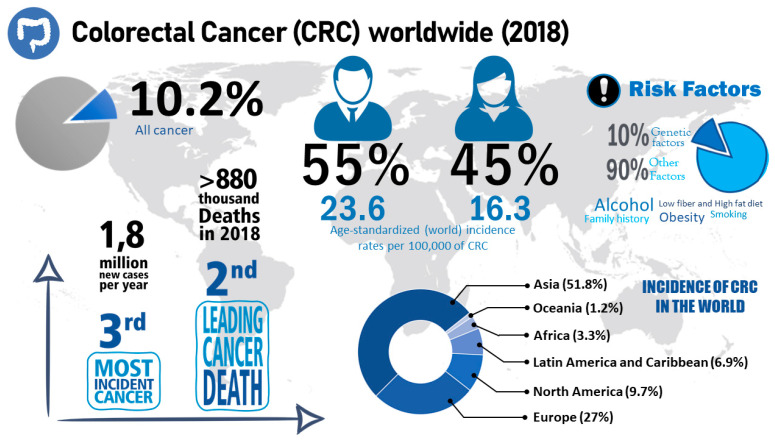
Characteristics of CRC. This figure presents data from CRC statistics [28], such as incidence, number of deaths and risk factors related to CRC.

**Figure 3 pharmaceutics-13-01321-f003:**
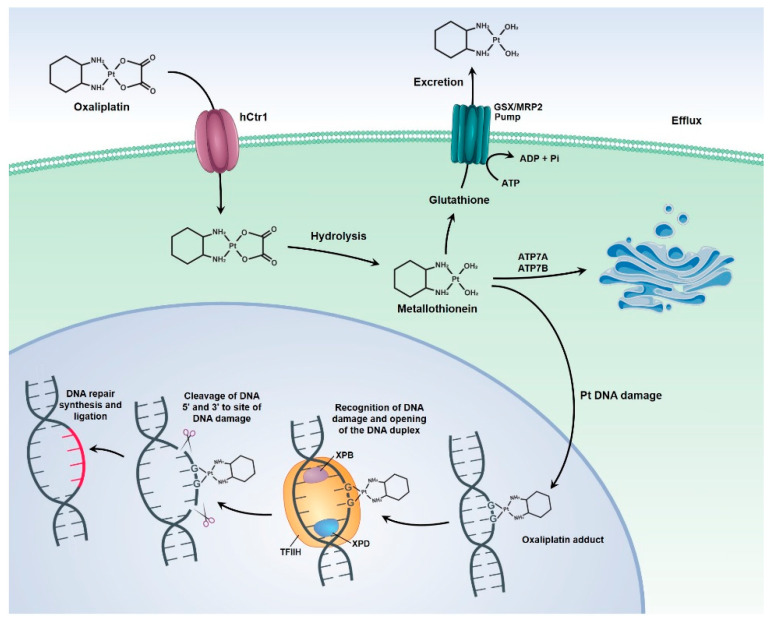
Tumor cell resistance mechanisms to OXA. The image shows the mechanisms of DNA repair and drug efflux.

**Figure 4 pharmaceutics-13-01321-f004:**
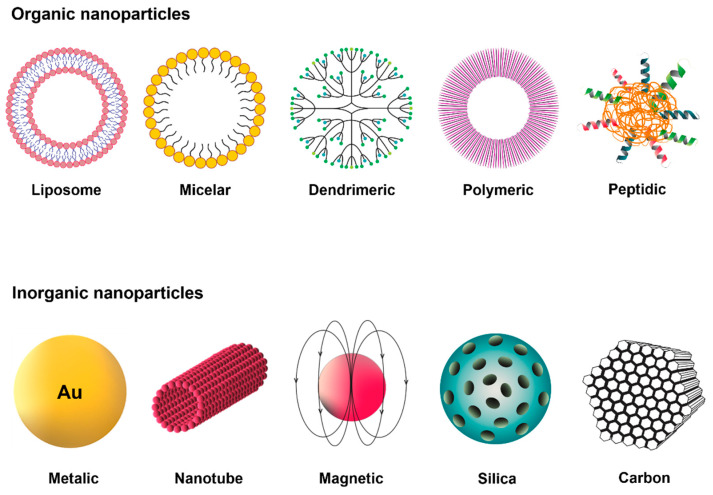
Different representations of the two large groups of NPs. These NPs are commonly used for biomedical applications and offer a significant potential as DDSs for cancer. They are divided into organic or inorganic, depending on the kind of material used for their composition.

**Figure 5 pharmaceutics-13-01321-f005:**
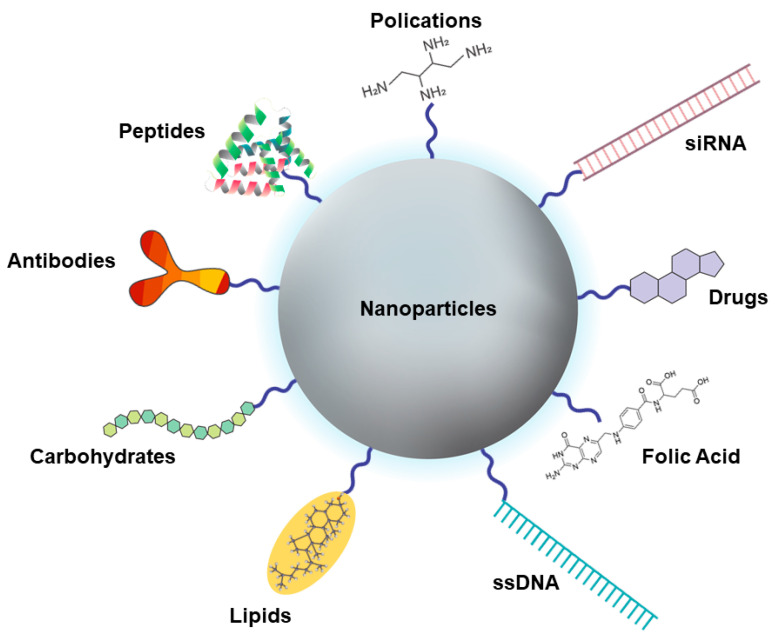
Delivery strategies for the active targeting of NPs. Graphical representation of surface-modified NPs with targeting molecules (antibodies, peptides, lipids and carbohydrates) for the delivery of cancer drugs.

**Table 1 pharmaceutics-13-01321-t001:** Most common treatment options based on the stage of colorectal cancer (excluding rectal cancer) using the TNM classification.

Stage of Cancer	Clinicopathological Characteristics	Treatment Modalities	Chemotherapeutic Agents	Targeted Therapy (Combined with Chemotherapy)
**Stage 1**	Primary tumor (T): Tis—carcinoma in situ; T1—tumor invades submucosa; T2—tumor invades muscularis propria; (N): N0—no regional lymph node metastasis; Distant metastasis (M): M0—no distant metastasis.	Local or surgical resection of malignant polyp or surgical procedure of tumor and local lymph nodes.	Not applied	Not applied
**Stage 2**	T: T3—tumor invades through muscularis propria into subserosal; T4—tumor directly invades other organs or structures, and/or perforates visceral peritoneum; N: N0—no regional lymph node metastasis; M: M0—no distant metastasis.	Surgical procedure without adjuvant chemotherapy. Adjuvant chemotherapy indicated in special cases, where high-risk characteristics are observed.	3–6 months of 5-flurouracil (FU) with leucovorin (LV), capecitabine or combination of 5-FU with LV and oxaliplatin (FOLFOX) or capecitabine and oxaliplatin (CAPOX).	Not applied
**Stage 3**	Any T; N: N1—metastasis in one to three regional lymph nodes; N2—metastasis in four or more regional lymph nodes; M: M0—no distant metastasis.	Surgery followed by adjuvant chemotherapy.	3–6 months of FOLFOX or CAPOX.	Not applied
**Stage 4**	Any T; Any N; M: M1—distant metastasis.	Radiotherapy, chemotherapy, immunotherapy, targeted therapies, palliative surgery/stenting, radiofrequency ablation, radio-embolization.	FOLFOIRI, FOLFIRI (5-FU, LV and irinotecan), FOLFOX, CAPIRI (capecitabine and irinotecan), CAPOX, 5-FU with LV, irinotecan, capecitabine and trifluridine plus tipiracil (Lonsurf).	Bevacizumab; cetuximab/panitumumab;

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
