# Peer review of "Nanocarriers as a Tool for the Treatment of Colorectal Cancer"

_pharmaceutics, 2021, doi:10.3390/pharmaceutics13081321_

Round 1
Reviewer 1 Report
The manuscript with title of " Nanocarriers as a tool for the treatment of colorectal cancer (CRC)". This is a topic that audience really care about, but has been comprehensively reviewed these years. Therefore, it’d be more attractive to the audience if the authors can focus on a particular issue regarding the use of specific type of nanocarriers (lipid-based or targeting nanocarriers etc.) Additionally, I found this manuscript is hard to follow as the structure is jumping. For instance, the authors introduced “nanoparticles as drug delivery systems” and followed with “Poly-lactic-co-glycolic acid”.
To improve the quality of this paper, I strongly suggest authors re-structure the manuscript before acceptance. The following is some of the issues for the authors to consider.
- A suitable introduction should be presented as too much information related to the pathways of carcinogenesis was include in the current content. A balanced view of the topic is highly recommended.
- Numbering of the sections should be corrected.
- Figure 2- a more updated information should be provided (if there is any).
- Please provide clinical trials based on the use of nanoparticles (NPs), particular for CRC treatments.
I suggest that the authors should address on the NPs used for CRC instead of general cancer.
Reviewer 2 Report
The manuscript entitled “Nanocarriers as a tool for the treatment of colorectal cancer” is well-written and provides an excellent overwiew about the use of nanotechnology in cancer, specially colorectal cancer. Nevertheless, some concerns should be adressed before being published in Pharmaceutics.
The authors should explain the carcinogenesis process. Please, define it.
Some typing errors should be corrected such as CAPEOX, CapOx.
The authors should provide some examples of hybrid nanocarries for the treatment of colorectal cancer.
“Micelles, which are spherical macromolecules formed by a hydrophilic outer layer and a hydrophobic interior” The authors should include in aqueous solution. In addition, micelles are not considered organic NPs. The authors should remove the concept of organic NPs for micelles in the manuscript. On the same way, micelles are usually characterized by having a size between 1 and 20 nm.
Liposomes can have a diameter of microns. Therefore, the reviewer does not understand when the authors affirm “not exceeding 400 nm”. Please, explain it in more detail.
The authors should mention different preparation methods of polymeric NPs.
Why have the authors affirmed that AuNPs have unique physical and chemical properties? Please, explain it.
“An example is the use of cetuximab or matuzumab on the surface of liposomal NPs”. The authors should remove agin the concept of liposomal NPs and use the term of liposomes.
Round 2
Reviewer 1 Report
Overall, the manuscript is well written. It is suitable for readers of Pharmaceutics, as it addresses an important issue of nanocarriers for colorectal cancer treatment, and I recommend the manuscript to be accepted in present form.
Author Response
1- Overall, the manuscript is well written. It is suitable for readers of Pharmaceutics, as it addresses an important issue of nanocarriers for colorectal cancer treatment, and I recommend the manuscript to be accepted in present form.
Answer: We thank the Reviewer very much for this positive feedback. We appreciate it.
Reviewer 2 Report
The authors should include the text in the manuscript about the different preparation methods of polymeric NPs.
The text "Polymer-lipid hybrid NPs (PLHNPs) are an example of hybrid nanocarriers for can-cer therapy, this nanosystem has a polymeric core, where the drug is encapsulated, sur-rounded by a lipid layer. The combination of polymeric NPs and liposomes, increases the biodistribution of the system, in addition to overcoming the limitations of each system would have alone[87-89]. Another example of a hybrid nanocarrier is a mesoporous silica NPs (MSNs) covalently bonded with poly (oligo(ethylene glycol) monomethyl ether methacrylate) (POEGMA) and targeting peptide (RDG) to perform system stabilization, and deliver in a targeted and effective manner the drug 5-fluorouracil (5-FU). This hybrid nanocarrier has better efficacy against CRC than the free drug, thus showing the im-portance of hybrid nanosystems and how they can be efficient in cancer therapy[90]." is repeated with different references in the manuscript.
"A recent study formulated and characterized liposome NPs containing irinotecan (IRI) to treat CRC; this research achieved a reduction in prostaglandins in colonic tissue in addi-tion to a reduction in tumors in mice, demonstrating promising results with these vesicles[97]." The authors continue to use the term of liposome NPs. Please, avoid it.
Author Response
1- The authors should include the text in the manuscript about the different preparation methods of polymeric NPs.
Answer: Thank you very much for point out our omission. We apologize for not adding the text of our reply to the manuscript. We corrected that error. You can find it in paragraph “4.1 Poly-lactic-co-glycolic acid”.
“Polymeric NPs consist of nanospheres or nanocapsules, and the preparation methods of such nanosystems vary depending on the drug to be encapsulated and the route of administration. However, there are two general methods of preparing polymeric NPs: the "top-down" and the "bottom-up" method, both methods used generate products that are obtained with aqueous colloidal suspensions. The steps of the technique that are used in the "top-down" include: emulsion evaporation, emulsion diffusion, coacervation and nanoprecipitation. The “bottom-up” method steps include: emulsion polymerization, interfacial polymerization, interfacial polycondensation and molecular inclusion. Thus, both use organic solvents to dissolve the polymer. Therefore, the solvent must be removed in subsequent reactions to avoid compound toxicity.”
2- The text "Polymer-lipid hybrid NPs (PLHNPs) are an example of hybrid nanocarriers for can-cer therapy, this nanosystem has a polymeric core, where the drug is encapsulated, sur-rounded by a lipid layer. The combination of polymeric NPs and liposomes, increases the biodistribution of the system, in addition to overcoming the limitations of each system would have alone[87-89]. Another example of a hybrid nanocarrier is a mesoporous silica NPs (MSNs) covalently bonded with poly (oligo(ethylene glycol) monomethyl ether methacrylate) (POEGMA) and targeting peptide (RDG) to perform system stabilization, and deliver in a targeted and effective manner the drug 5-fluorouracil (5-FU). This hybrid nanocarrier has better efficacy against CRC than the free drug, thus showing the im-portance of hybrid nanosystems and how they can be efficient in cancer therapy." is repeated with different references in the manuscript.
Answer: Thank you very much for pointing this out. We apologize for our mistake and removed the text in paragraph “4.1 Poly-lactic-co-glycolic acid”. In this place the text of comment 1 was the text that should have been inserted. With the correction of the text, the references are correct now.
3- "A recent study formulated and characterized liposome NPs containing irinotecan (IRI) to treat CRC; this research achieved a reduction in prostaglandins in colonic tissue in addi-tion to a reduction in tumors in mice, demonstrating promising results with these vesicles[97]." The authors continue to use the term of liposome NPs. Please, avoid it.
Answer: Thank you very much for pointing this out. We apologize for our omission and changed the text accordingly. In addition, we searched our manuscript for the term to possibly find other omissions, but we did not find any. The term “liposome NPs” is not contained in our manuscript anymore. We hope that the Reviewer is happy with our adaptations and changes and will consider the manuscript for publication in its current state.